# Isoquercitrin Attenuates Osteogenic Injury in MC3T3 Osteoblastic Cells and the Zebrafish Model via the Keap1-Nrf2-ARE Pathway

**DOI:** 10.3390/molecules27113459

**Published:** 2022-05-27

**Authors:** Xue Li, Dongyue Zhou, Di Yang, Yunhua Fu, Xingyu Tao, Xuan Hu, Yulin Dai, Hao Yue

**Affiliations:** Jilin Ginseng Academy, Key Laboratory of Active Substances and Biological Mechanisms of Ginseng Efficacy, Ministry of Education, Changchun University of Chinese Medicine, Changchun 130117, China; jokerlee0910@163.com (X.L.); zhoudongyue1203@163.com (D.Z.); yangdi8878@163.com (D.Y.); fuyunhua1998@163.com (Y.F.); xyutao2021@163.com (X.T.); dylan6s@foxmail.com (X.H.)

**Keywords:** isoquercitrin, osteogenic development, antioxidant

## Abstract

Isoquercitrin (IQ) widely exists in natural products, with a variety of pharmacological activities. In this study, the anti-apoptotic and antioxidative activities of IQ were evaluated. IQ showed protective activity against 2, 2′-azobis [2-methylpropionamidine] dihydrochloride (AAPH)-induced cell damage, as well as a marked reduction in reactive oxygen species (ROS). The evidence of IQ regulating Keap1-Nrf2-ARE and the mitochondrial-mediated Caspase 3 pathway were found in the MC3T3 osteoblastic cell line. Furthermore, IQ significantly decreased ROS production, apoptosis, and lipid peroxidation in AAPH-treated 72 h post-fertilization (hpf) zebrafish, as observed via DCFH-DA, acridine orange (AO), and a 1,3-bis(diphenylphosphino) propane (DPPP) probe, respectively. In AAPH-treated 9 day post-fertilization (dpf) zebrafish, IQ strongly promoted osteogenic development, with increased concentrations by calcein staining, compared with the untreated group. In a molecular docking assay, among all signal proteins, Keap1 showed the strongest affinity with IQ at −8.6 kcal/mol, which might be the reason why IQ regulated the Keap1-Nrf2-ARE pathway in vitro and in vivo. These results indicated that IQ promotes bone development and repairs bone injury, which is valuable for the prevention and treatment of bone diseases.

## 1. Introduction

Bone is a dynamic tissue. Bone development is not confined to adolescence, but also constantly changes in adults [1]. The growth, development, metabolism, and aging of bone tissue are manifested as the increase or decrease in bone mass [2]. The dynamic balance of bone remodeling and bone metabolism is carried out histologically, including bone formation by osteoblasts, bone absorption by osteoclasts, and the differentiation, proliferation, apoptosis, and transformation of osteoblasts [3]. Skeletal homeostasis imbalance ultimately leads to bone dysplasia and bone-related diseases [4]. Bone-related diseases, including common ailments, such as fractures, osteoporosis, and osteoarthritis [5], affect a large number of individuals. With the existing prevention and treatment methods, the incidence and mortality of bone-related diseases are still gradually increasing [6], which has brought a huge financial burden to societies worldwide. In addition to genetic factors, the steady-state imbalance of osteoblasts and osteoclasts can directly cause bone dysplasia and regeneration disorders, which becomes a direct cause of many bone diseases [7]. Thus, new and complementary treatments for bone development and injury repair must be developed to prevent the occurrence of bone-related diseases, delay their development, or reverse the injuries caused by such diseases.

IQ is a flavonol compound and is a natural active constituent that widely exists in many traditional Chinese medical herbs, such as *Amomum villosum Lour* and *Morus alba Linn* [8]. In recent years, IQ has attracted the attention of many scholars because of its antioxidant and anti-osteoporosis properties [9]. IQs have extensive antioxidant activity, which is mainly attributed to their ability to activate nuclear factor erythroid 2-related factor (Nrf2). Nrf2 is an important cell protective transcription factor responsible for encoding intracellular antioxidants, such as heme oxygenase-1 (HO-1) and NADH-quinone-oxidoreductase-1 (NQO-1). The activity of Nrf2 is closely related to Kelch-like ECH-related protein 1(Keap1). Under non-stress conditions, Keap1 and Nrf2 form a complex to inhibit the activity of Nrf2. Eventually, Nrf2 is degraded by protease in the cytoplasm. However, under stress, Keap1 is degraded after oxidation or covalent modification by cysteine residues, allowing Nrf2 to enter the nucleus and accumulate, thus encoding the downstream antioxidant response components and exerting antioxidant activity [10]. A previous study has found that IQ could promote the proliferation and differentiation of osteoblasts and bone marrow mesenchymal stem cells, indicating its broad application potential in bone development, bone injury repair, and bone-related diseases [11]. Based on previous reports, IQ has anti-osteoporotic activity, but it has not been verified by animal experiments [12]. The study of its mechanism is not precisely elucidated, which has been limited to the research on the RUNX2 or BMP pathway [13]. Therefore, research on the activity and mechanism of IQ in promoting bone development is important.

In general, oxidative stress is due to the imbalance between oxidation and antioxidation in the body, resulting in a large number of oxidative intermediates, which lead to aging and diseases [14]. In recent years, many studies have proven the close relationship between oxidative stress and osteogenesis development. Increased levels of oxidative stress, especially reactive oxygen species, can aggrandize the absorption capacity of osteoclasts and disrupt the balance between bone development and bone absorption, leading to diseases related to bone developmental defects and bone damage [15]. Cycloastragenol inhibits the formation and function of osteoclasts by regulating RANKL-induced Keap1-Nrf2-ARE signaling pathways [16]. AAPH, as a free radical inducer, is used to induce the oxidative damage of nucleus pulposus cells to study intervertebral disc degeneration spinal degenerative diseases [17]. In this study, MC3T3-E1 mouse embryonic osteoblast precursor cells (MC3T3) and zebrafish skeleton injury models were induced by AAPH to explore the bone injury repair effect and related protein pathways of IQ.

The function of osteoblasts and osteoclasts in the process of zebrafish bone development is similar to that of human bone biomineralization [18]. The initial spine of zebrafish is formed at 7 dpf, and it gradually extends from the back to the abdomen [19]. Given the visual characteristics of larvae, tissue staining of the spine can track bone mineralization and bone development. Therefore, zebrafish were selected as the model organism to investigate bone diseases.

This study aimed to assess the repair effect of IQ on AAPH oxidative injury-induced bone dysplasia by detecting the MC3T3 cell apoptosis, oxidative stress markers, and bone mineralization in zebrafish and to verify whether it works through the Keap1-Nrf2-ARE antioxidant stress pathway and the caspase apoptosis pathway from the protein expression level. The result of this study will provide valuable information for the application of natural phytoestrogen IQ in bone development and bone diseases.

## 2. Results

### 2.1. In Vitro Qntioxidant Test Results

#### 2.1.1. Screening of IQ and AAPH Concentration in MC3T3 Cells

An MTT assay was used to observe the effects of IQ with different concentration gradients on the viability of MC3T3 cells (Figure 1a). The cell viability of MC3T3 cells treated with the maximum concentration of IQ (100 μM) was over 80%, indicating that IQ was nontoxic to MC3T3 cells within this concentration gradient and could be used for further activity experiments. The solvent used in the cell experiment was phosphate buffer. The same method was used to screen the concentration of AAPH induced by oxidative injury of MC3T3 (Figure 1b). When the concentration was 10 mM AAPH, the cell viability was 60%. Therefore, 10 mM of AAPH was selected as the concentration of the inducer for the MC3T3 cell oxidative damage model in the subsequent experiment.

#### 2.1.2. Determination of the Intracellular ROS Level in MC3T3 Cells

The relative levels of ROS in MC3T3 cells in the CON group, AAPH group, and IQ-AAPH group (12.5, 25, 50, and 100 μM) were measured (Figure 1c). Under the induction of AAPH oxidation, with the elevation of IQ concentration, the ROS level in MC3T3 cells reduced distinctly. Compared with the AAPH group, the relative levels of ROS in the IQ-AAPH group with three concentration gradients showed visible discrepancies. An evident difference was observed with the increase in IQ concentration.

#### 2.1.3. Determination of MC3T3 Cell Viability

The absorbance of the CON group, AAPH group, and IQ-AAPH group (12.5, 25, 50, and 100 μM) was measured (Figure 1d). Compared with AAPH, the cell viability of the 25 μM IQ-AAPH group showed a distinct difference, and the cell viability of the 50 μM IQ-AAPH group and the 100 μM IQ-AAPH group showed an extremely significant dissimilarity. Furthermore, the cell viability showed IQ concentration dependence.

#### 2.1.4. MC3T3 Cell Apoptosis Staining Experiment

Hoechst 33,342 staining was used to detect the formation of apoptotic bodies in MC3T3 cells (Figure 1e). Compared with the CON group, the blue fluorescence intensity of the AAPH group was evidently enhanced. With the magnification of IQ concentration, the blue fluorescence intensity gradually decreased, indicating that IQ had a repairing effect on the apoptosis of MC3T3 cells induced by AAPH. The anti-apoptotic activity was dose dependent.

AO/EB staining was used to detect the apoptotic stage of MC3T3 cells (Figure 1f). The cells in the CON group showed uniform green fluorescence, whereas the green granular fluorescence and orange fluorescence appeared in the AAPH group, and the total cell volume was also distinctly reduced. With the increasing concentration of IQ, the relative intensity of orange fluorescence was gradually receded, and the cell volume was also magnified, demonstrating that IQ improved early apoptosis, late apoptosis, and nonapoptotic death of AAPH-induced MC3T3 cells, and the more significant the activity.

#### 2.1.5. Cell Apoptosis Cycle Detection

Based on the results of the above mentioned cell experiments, the IQ-AAPH groups with different concentrations were reduced to a group of only 100 μM, and the cell apoptosis was evaluated by flow cytometry (Figure 2). Compared with the CON group, the proportion of viable cells (lower left quadrant, R1) in the AAPH group was visibly abated, whereas the proportion of early apoptotic cells (lower right quadrant, R2) and late apoptotic cells (upper right quadrant, R3) were evidently increased. After the addition of 100 μM of IQ, the number of living cells was distinctly magnified, and the number of early apoptotic cells and late apoptotic cells significantly decreased.

#### 2.1.6. IQ-Regulated Antioxidant and Apoptosis Pathways

The results showed that the expression of Keap1 protein in the AAPH group was downregulated compared with that in the CON group. Under 100 μM IQ treatment, the expression of Keap1 protein was observably increased. After AAPH treatment, the protein expression of HO-1 and NQO-1 was attenuated, and it was evidently advanced after 100 μM IQ treatment (Figure 3a). The results showed that the expression level of Bcl-2 protein in the CON group was high, but it was obviously downregulated in the AAPH group, and the protein expression level was dramatically upregulated after 100 μM IQ treatment. Compared with the CON group, the expression of Bax, CytoC, and Caspase3 proteins in the AAPH group was modified, and it was evidently mitigated under 100 μM IQ treatment (Figure 3b).

### 2.2. Effect of IQ on Antioxidation and Development of Osteogenesis in Zebrafish

#### 2.2.1. Antioxidant Effect of IQ on AAPH-Treated Zebrafish Embryos (72 hpf)

The 12.5 μM IQ-AAPH group showed few evident therapeutic effects compared with the AAPH group in the cell experiment; thus, the concentration gradient in the IQ-AAPH group was adjusted to 25, 50, and 100 μM in the subsequent in vitro experiments.

Based on the screening results of AAPH concentration, 13 mM of AAPH was selected as the inducer of oxidative damage in 72 hpf zebrafish (Appendix A). The solvent used in the zebrafish experiment was double distilled water. The ROS levels in the CON group, AAPH group, and IQ-AAPH group (25, 50, and 100 μM) were stained with a DCFH-DA staining agent (Figure 4a). The green fluorescence intensities of larvae in the AAPH group were augmented, whereas those of the IQ-AAPH group (25, 50, and 100 μM) were gradually decreased. The apoptotic cells in five groups of zebrafish were stained with AO dye (Figure 4b). Compared with the CON group, the green fluorescence intensity of the larvae AAPH group was evidently enhanced. With the gradual increase in IQ concentration, the green fluorescence intensity gradually weakened. DPPP staining was used to stain the lipid peroxidation levels of zebrafish in the five groups (Figure 4c). The blue fluorescence intensity of larvae fish in the AAPH group was evidently strengthened compared with the CON group, and the blue fluorescence intensity of the IQ-AAPH groups (25, 50, and 100 μM) was significantly diminished compared with the AAPH group. The heart rate of 72 hpf zebrafish was monitored (Figure 4d). The heart rate of larvae in the AAPH group was significantly higher than that in CON group. After IQ treatment at different concentrations, the heart rate decreased significantly.

#### 2.2.2. Effect of IQ on Repairing Bone Damage of AAPH-Treated Zebrafish (9 dpf)

Based on the screening results of AAPH concentration, 2 mM of AAPH was selected as the inducer of the 9 dpf zebrafish bone injury model (Appendix A). The 9 dpf zebrafish were stained with calcein (Figure 4e). Compared with the AAPH group, the CON group zebrafish showed bright green fluorescence in the spinal position, whereas the intensity of green fluorescence in the spine of IQ groups with different concentrations increased, and the fluorescence intensity of the 50 μM IQ group and the 100 μM group displayed an evident difference (Figure 4f).

### 2.3. Molecular Docking

In this study, Keap1, NQO-1, HO-1, Bcl-2, Bax, Caspase3, and IQ were docked in reverse molecular docking. When the binding energy is less than 0, ligands and receptors can bind freely [20]—the lower the energy, the stronger the binding ability between them. The conformation of IQ with binding energy less than −7 was screened (Table 1, Figure 5), and Keap1 protein (PDB ID 1ZGK) showed a strong affinity to IQ.

## 3. Discussion

The bone pedigree includes osteoblasts, osteocytes, and chondrocytes, which maintain and repair bones during the dynamic balance and injury of bone and cartilage [21]. Osteoblasts are responsible for bone formation, which secrete type I collagen, osteopontin, osteocalcin, and alkaline phosphatase, which provides structural support for bones by mineralizing calcium deposited in the form of hydroxyapatite, along with type I collagen [22]. Bone injury is repaired through the changing microenvironment to promote osteoblast differentiation into osteoblasts and to repair damaged tissue [23].

ROS and free radicals are necessary for cellular signaling and other physiological functions [24]. However, excessive ROS inhibits the proliferation and differentiation of osteoblasts and generates apoptosis, thereby hindering bone mineralization and osteogenesis. ROS can activate the proliferation and differentiation of osteoclasts and promote bone resorption [25]. The hysteresis of the bone formation rate behind the rate of bone resorption indicates the imbalance of bone remodeling, which leads to a series of diseases, including bone dysplasia [26]. Therefore, in this study, the oxidative damage of MC3T3 cells generated by AAPH was selected as the cellular model. The results of intracellular ROS relative level detection showed that IQ had an inhibitory effect on AAPH-induced ROS in MC3T3 in a dose-dependent manner. Hence, IQ may protect MC3T3 cells by inhibiting excessive intracellular ROS, thereby promoting bone development. Osteoblast apoptosis can directly hinder the function of bone mineralization and regeneration [27]. The effect of cell viability indicated that IQ had a therapeutic influence on MC3T3 cell apoptosis caused by AAPH-caused oxidative damage, and the therapeutic impact was evident when the concentration of IQ was 100 μM. The outcomes of flow cytometry indicated that the early and late apoptotic cells in the IQ-AAPH group evidently decreased compared with the AAPH group, and the number of living cells increased significantly. Furthermore, IQ may promote osteogenesis by inhibiting ROS production and apoptosis in MC3T3 cells.

Keap1-Nrf2-ARE and the caspase pathway have regulatory effects on cellular oxidative damage and apoptosis. Therefore, the protein-level mechanism of IQ on bone damage repair was explored from the perspective of the protection and damage mechanism. The Keap1-Nrf2-ARE signaling pathway is a vital mechanism in cells for protection against oxidative stress injury [28]. Under normal physiological conditions, the main role of Keap1 is to degrade Nrf2 ubiquitin proteases. During oxidative stress, Keap1 is inactivated, and the ubiquitination of Nrf2 stops, resulting in the activation of downstream pathways and the expression of HO-1 and NQO-1 proteins [29]. The protein expression of Keap1, HO-1, and NQO-1 in the CON group, AAPH group, and IQ-AAPH group showed that the plerosis function of IQ on AAPH oxidative damage may be achieved by regulating the Keap1-Nrf2-ARE pathway.

Apoptosis (also known as programmed cell death) is regulated by a complex signal network. The mitochondria-mediated caspase enzyme activation pathway is the main apoptotic pathway, primarily by Bcl-2 family proteins acting on the mitochondria, promoting the release of CytoC into the cytoplasm, activating Caspase3, and inducing apoptosis [30]. In this study, we monitored the expression of several representative proteins in this pathway. Bcl-2 and Bax affect the state of cells by controlling the permeability of the mitochondrial membrane to regulate the release of apoptosis activators such as CytoC [31]. Bax dimer opened channels on the membrane to increase permeability, whereas Bcl-2 and Bax formed isomers to reduce permeability. The results revealed that IQ has an anti-apoptotic effect on AAPH-stimulated MC3T3 cells, which might be achieved by regulating the mitochondria-mediated Caspase3 protein pathway.

Zebrafish have many advantages for research, including strong fecundity, external fertilization, rapid embryonic development, and visualization of larvae. Thus, the zebrafish has become an effective alternative vertebrate model for the study of human bone diseases [32]. The preliminary detection of the oxidative stress level of 72 hpf zebrafish is not only evaluated directly by measuring ROS, but also combined with the indirect detection of biomolecular damage markers because of the instantaneous characteristics of ROS [33]. Direct staining of ROS, apoptotic cells, and lipid peroxidation in 72 hpf zebrafish demonstrated that IQ could restrain ROS, apoptotic cells, and lipid peroxidation in 72 hpf zebrafish damaged by AAPH, and the effect was evident when the concentration of IQ was 100 μM. This result, combined with the observations of zebrafish heart rate, further confirmed that IQ has a therapeutic function on AAPH-induced oxidative stress and apoptosis.

The zebrafish is a bony fish, and its bone development is basically consistent with the biomineralization of human bone. Zebrafish have cell types similar to those of the mammalian bone systems, and they also have osteoblasts, osteoclasts, and bone cells. In addition, the function of these cells in zebrafish bone development is similar to that of mammals [34]. With regard to the shape of the spine, the physiological curvature and number of the vertebrae of zebrafish are similar to those of humans (30 to 32 zebrafish and 33 in humans). The initial spine of the larvae formed on 7 dpf begins in the area of the back and extends to the abdomen. The 3rd, 4th, 5th, and 6th spinal cones are formed first, followed by the 1st and 2nd vertebrae [35]. Tissue staining of the spine can track bone mineralization and bone development [36]. Calcein staining of 9 dpf zebrafish showed that after continuous oxidative stimulation of AAPH, the green fluorescence intensity of zebrafish in the AAPH group was faint, indicating that oxidative stress hindered zebrafish bone mineralization. After IQ treatment, the green fluorescence intensity augmented gradually, indicating that IQ promoted calcium deposition and bone development. This result further confirmed that IQ can rehabilitate bone dysplasia and AAPH-induced oxidative damage at the animal level. By comparing the damage of zebrafish at 72 hpf and 9 dpf, it can also be speculated that the accumulation of oxidative damage will cause zebrafish osteogenesis defects, and the specific mechanism and relationship between the two require further experiments to prove.

Through the reverse molecular docking of Nrf2-Keap1-ARE and mitochondria-mediated caspase signal pathway with IQ, the results indicated that Keap1 and IQ showed a strong affinity. Keap1 belongs to the upstream protein of the Nrf2-Keap1-ARE antioxidant stress signal pathway. By participating in the polymerization and depolymerization of Nrf2 proteins to regulate the expression trend of downstream protein, IQ may activate downstream HO-1 by binding to the Keap1 of this pathway, NQO1, and other protein expressions, thereby promoting osteogenic development and injury repair. It has been reported in previous studies that IQ rescued HT22 mouse hippocampal neuronal cells from glutamate (Glu)-induced oxidative cell death by the Nrf2/HO-1 pathway [37]. These results indicated that the protective effects of IQ on cell death and apoptosis induced by oxidative damage caused by different reagents might be regulated by the same protein pathway. The docking results show that if isoquercitrin can freely enter cells, differences in cell membrane penetration or cellular uptake may be the possible factors explaining the differential protection by quercetin derivatives on different types of cells.

## 4. Materials and Methods

### 4.1. Reagents and Instruments

MC3T3 cells are commercially available from the American Type Culture Collection. Minimum eagle’s medium (MEM), penicillin/streptomycin solution (P/S), tyrisin (Hyclone, Utah, USA), fetal bovine serum (FBS; Clark, VA, USA), phosphate-buffered solution (PBS; Leagene, Beijing, China), BCA protein assay reagent kit, Hoechst 33342, acridine orange (AO), 3-(4,5)-dimethylthiahiazo(-z-y1)-2,5-diphenytetrazoliumromide (MTT; Solarbio, Beijing, China), IQ, 2′,7′-dichlorodihydrofluorescein diacetate (DCFH-DA), acridine orange/ethidium bromide (AO/EB; Yuanye, Shanghai, China), 2,2′-azobis [2-methylpropionamidine] dihydrochloride (AAPH; Macklin, Shanghai, China), Annexin V-FITC/PI Apoptosis Detection Kit (Becton Dickinson, Franklin Lake, NJ, USA), 1,3-bis(diphenylphosphino) propane (DPPP; TCI, Tokyo, Japan), sodium dodecyl sulfate polyacrylamide gel electrophoresis preparation kit (SDS-PAGE), electrochemiluminescence (ECL) substrate kit (Biosharp, Beijing, China), β-actin, Nrf2, Keap1, NQO-1, HO-1, Caspase3, Bax, Bcl-2, cytochrome c (CytoC), IgG/AP-goat anti-rabbit IgG (H+L; Proteintech, Rosemont, IL, USA), polyvinylidene fluoride (PVDF) membrane (Merck, Darmstadt, Germany), fluorescence microscope (Nikon, Tokyo, Japan), CO_2_ incubator (Thermo, Waltham, MA, USA), infinite M200 pro multifunctional microplate reader (Tecan, Männedorf, Switzerland), electrophoresis apparatus (BIO-RAD, Hercules, CA, USA), and amnis flow sight (Luminex, Austin, TX, USA) were used in this study.

### 4.2. Cell Lines and Cell Culture

MC3T3 osteoblastic cells were maintained by supplementing MEM with heat-inactivated 10% FBS and 1% P/S in a humidified atmosphere containing 5% CO_2_ at 37 °C. The cells were passaged every 3 days with 0.25% trypsin and PBS at a dilution ratio of 1:4.

### 4.3. Animals

Zebrafish, a wild-type AB strain, were purchased from Nanjing EzeRinka Biotechnology Co., Ltd., and maintained and treated in accordance with the guidelines of the zebrafish model biological database (http://zfin.org, 3 April 2020). They were exposed to 14 h light and 10 h darkness to maintain circadian rhythms. Fertilization occurred by natural oviposition at the beginning of the photoperiod. The embryos were collected and raised in E3 water at 28 °C. E3 water was made of magnesium chloride, calcium chloride, potassium chloride, and sodium chloride.

### 4.4. Antioxidant Effect of IQ on MC3T3 Cells

#### 4.4.1. Screening of AAPH Concentration in MC3T3 Cells

The exponentially growing MC3T3 cells (1 × 10^4^ cells per well) were inoculated in 96 well plates and incubated at 37 °C with 5% CO_2_ until the density reached 80%. PBS was added to the CON group, and inducers (2.5, 5, 7.5, 10, and 15 mM) at different concentrations were added to the AAPH group. After incubation for 23 h, 50 μL of freshly prepared MTT (0.5 mg/mL) was added. After incubation for 3 h, the supernatant was discarded, and 200 μL of DMSO was added to each well and placed on the shaking bed for 3 h. The optical density (OD) value was detected by using a microplate reader at a wavelength of 540 nm. A survival rate of 60% was used as the concentration of the inducer for the MC3T3 oxidative stress model.

#### 4.4.2. Screening of IQ Concentration

The AAPH group was stimulated with different concentrations of IQ (12.5, 25, 50, and 100 μM) for 24 h, and the control (CON) group was given the same amount of E3 water. The other operations were the same. Cytotoxicity was selected in accordance with the operation presented in Section 4.4.1.

#### 4.4.3. Detection of Reactive Oxygen Species (ROS) Levels in MC3T3 Cells

Oxidative stress can be directly assessed by measuring ROS. For the CON, AAPH, and IQ-AAPH groups (12.5, 25, 50, and 100 μM), the operation was the same as that shown in 4.4.1. When the cell density reached 80%, the drug was applied for 1 h, followed by the addition of 10 mM of AAPH, and the incubation was continued for 3 h. The 10 μL DCFH-DA fluorescent probe was added to each well. The OD value was detected immediately, at an excitation wavelength of 480 nm and an emission wavelength of 530 nm, and the relative level of ROS was calculated.

#### 4.4.4. Determination of the Viability of MC3T3 Cells

The experimental grouping was consistent with that shown in Section 4.4.3, and the operation was the same as that shown in Section 4.4.1. When the cell density reached 80%, the drug was given; 10 mM of AAPH was added after 1 h; the incubation was continued for 23 h, and the cell survival rate was detected by MTT assay.

#### 4.4.5. Hoechst 33,342 Staining

The formation of apoptotic bodies in cells was detected by Hoechst 33,342 staining to evaluate apoptosis. The cells were seeded on a 12-well plate and cultured until the cell volume reached 80%. After treatment, 25 μL of different concentrations of IQ (2.5, 5, 7.5, 10, and 15 mM) were added to each well for culture for 1 h, and then 25 μL of AAPH (10 mM) were added for continuous culture for 23 h. The culture solution was discarded and rinsed two times with PBS. Subsequently, 130 μL of PBS were added to each well. After being added with 15 μL Hoechst 33342, the mixture was gently shaken and homogenized. After incubation in the dark for 5 min at room temperature, the stained cells were im-mediately observed under a fluorescence microscope.

#### 4.4.6. AO/EB Staining

AO/EB staining was used to detect the apoptosis period of MC3T3, as well as viable cells (nuclear chromatin showed green uniform fluorescence), early apoptotic cells (nuclear chromatin showed green granular fluorescence), late apoptotic cells (nuclear chromatin showed orange-red granular fluorescence), and non-apoptotic dead cells (nuclear chromatin showed orange-red uniform fluorescence). Cell plating and treatment are consistent with the methods shown in Section 4.4.5. After being washed two times with PBS, 196 μL of PBS were added into each well, followed by the addition of 2 μL of AO staining solution and 2 μL of EB staining solution. After being mixed evenly, the wells were incubated for 5 min at room temperature in the dark. The stained cells were observed under a fluorescence microscope.

#### 4.4.7. Apoptosis Analysis by Flow Cytometry

MC3T3 cells in the exponential growth phase were uniformly inoculated into 6-well plates (2 × 10^5^ cells per well) and divided into three groups: the CON group, AAPH group, and IQ-AAPH group. They were incubated at 37 °C with 5% CO_2_. When the cell density reached 80%, PBS was added to the CON group; 10 mM of AAPH were added to the AAPH group, and 10 mM of AAPH and 50 μg of IQ were added to the IQ-AAPH group. Incubation was continued for 24 h. Afterward, the culture solution was discarded; the suspended dead cells were washed with precooled PBS; trypsin was added for digestion for 1 min. Next, an equal amount of the culture solution was added to stop the digestion, and then the cell suspension was transferred into a centrifuge tube and immediately centrifuged at 4 °C (1000 rpm for 3 min). Afterward, the supernatant was discarded, and then 1 mL of precooled PBS was added for washing 2 times. Finally, 500 μL of resuspension solution were added into the final precipitate for air suspension. Separately, 200 μL of cell suspension in the CON group, AAPH group, and IQ-AAPH group were added to 1.5 mL ep tube, into which 5 μL of FITC Annexin V and 5 μL of PI staining solution were added. Separately, 100 μL of cell suspension in the CON group were used for PI staining and FITC Annexin V staining. After being incubated in the dark for 15 min, the cells were detected by flow cytometry, and data were analyzed by IDEAS.

### 4.5. Western Blot Assay

#### 4.5.1. Cell Protein Sample Collection

The cells were divided into the CON group, AAPH group, and IQ-AAPH group. The MC3T3 cells in the exponential growth phase were inoculated with 2 × 10^5^ cells per well in a 60 mm diameter culture dish and incubated at 37 °C with 5% CO_2_ until the density reached 80%. The three groups were treated with PBS, 12 mM of AAPH solution, and 100 μM of AAPH solution and incubated for 24 h, and the culture medium was discarded. After the suspended dead cells were washed with PBS and digested with trypsin, the obtained cell suspension was immediately centrifuged at 4 °C (12,000 rpm for 3 min), and then the trypsin was discarded. The residual trypsin was washed with PBS; the obtained cell precipitate was centrifuged at 4 °C (12,000 rpm for 3 min), and then 100 μL of strong lysate were added for low-temperature lysis for 1 h, wherein the cells were vortexed for 5 s every 10 min. After the cells were centrifuged at 4 °C (12,000 rpm for 15 min), the obtained supernatant was MC3T3 cell protein.

#### 4.5.2. Determination of Protein Concentration and Western Blot Experiment

The protein concentration of the obtained MC3T3 cell protein sample was measured by the BCA kit. The protein sample was quantified as 15 μg per well. The protein was separated by 10% SDS-PAGE gel electrophoresis and then transferred to the PVDF membrane. The successfully transferred membrane was inspected by Ponceau red staining. The staining agent on the PVDF membrane was cleaned by tris HCl buffer salt solution with tween (TBST), which was blocked by 5% defatted milk at room temperature for 3 h. The corresponding primary antibody (1:1000) was added and incubated at 4 °C overnight. TBST was used to wash the PVDF membrane, and the secondary antibody was incubated at room temperature for 2.5 h. TBST was used to wash the PVDF membrane again, and a high sensitivity ECL solution was used to develop a chemiluminescence apparatus for imaging. The experimental results were analyzed by Image J for the gray values of protein bands.

### 4.6. In Vivo Antioxidant Activity Detection of IQ

#### 4.6.1. ROS Staining Experiment of 72 hpf Zebrafish

The AAPH concentration was screened to induce oxidative damage of 72 hpf zebrafish. The 6 hpf embryos were randomly divided into six groups: a CON group and five AAPH groups (10, 12, 13, 14, and 15 mM). Fifteen embryos were allocated in each group. The survival of the embryos was recorded every 6 h, and dead embryos were removed in time. After 72 h stimulation, the survival rate of larvae was calculated. The concentration of AAPH with a survival rate of 80% was selected as the concentration of the inducer for subsequent experiments.

The 6 hpf zebrafish embryos were selected and randomly divided into five groups: the CON group, AAPH group, and three IQ-AAPH groups (25, 50, and 100 μM). Fifteen embryos in each group were placed in a 12-well plate and dosed at 8 hpf; 13 mM of AAPH were added at 9 hpf, and the water was changed once a day. After continuous dosing and oxidation induction treatment of 72 hpf, a DCFH-DA staining agent (20 μg/mL) was added for staining, followed by incubation in the dark for 2 h, washing out the staining solution, and photographic observation under a blue fluorescence microscope.

#### 4.6.2. Apoptotic Cell Staining Experiment of 72 hpf Zebrafish

The grouping and dosing method was consistent with that shown in Section 4.6.1. After 72 hpf, an AO staining agent (5 μg/mL) was added for staining, and the samples were incubated in the dark for 1.5 h, after which the staining solution was washed away, and the samples were photographed under the blue fluorescence microscope for observation.

#### 4.6.3. Lipid Peroxidation Staining Experiment of 72 hpf Zebrafish

Given the transient nature of ROS, indirect methods were often used to assess oxidative stress levels. In this study, lipid peroxidation levels in 72 hpf zebrafish were assessed. Grouping and administration methods were consistent with those shown in Section 4.6.1. After 72 hpf, samples were stained with a DPPP agent (25 μg/mL) and incubated in the dark for 1 h, and the staining solution was washed, and the samples were photographed under a green fluorescence microscope.

#### 4.6.4. Determination of Zebrafish Heart Rate

Heart rate is also one of the indicators for evaluating the level of oxidative stress in vivo. The heart rate of 72 hpf zebrafish was detected; grouping and administration methods were consistent with those shown in Section 4.6.1. After 72 hpf, the zebrafish heartbeats per minute were counted under a microscope to calculate an average value.

### 4.7. Effect of IQ on Bone Injury Repair of Zebrafish

#### 4.7.1. AAPH Concentration Screening of 9 dpf Zebrafish

The AAPH concentration was screened to induce osteoporosis in 9 dpf zebrafish: 3 dpf larvae were randomly divided into six groups with 30 embryos in each group. Starting from 5 dpf, larvae were fed and given different concentrations of AAPH solution to 9 dpf (1, 2, 5, 8, 10, and 13 mM), and the dead larvae were removed in time. Then, the survival rate was recorded. The survival rate was 80%, which served as the concentration of the oxidative stress inducer for larvae.

#### 4.7.2. Staining Experiment of 9 dpf Zebrafish Osteogenesis

The 3 dpf larvae were randomly divided into five groups: the CON group, AAPH group, and IQ-AAPH group (25, 50, and 100 μM). Thirty larvae in each group were fed at 5 dpf, once in the morning and once in the evening. The medium was changed 1 h after each feeding, and drugs were added after each water change. Then, 2 mM of AAPH were added 1 h after the addition of the drug. The larvae were fed continuously to 9 dpf, and the culture medium was discarded. Two milliliters of calcein staining agent (2 mg/mL) were added to each well for 5 min in the dark, and the staining agent was removed. Spine development was observed under a blue fluorescence microscope.

### 4.8. Reverse Molecular Docking of IQ

AutoDockTools was used to dock IQ with proteins on the Nrf2-Keap1-ARE and mitochondria-mediated caspase signal pathways to study the interaction between IQ and protein molecules. The standardized human protein target information was obtained from the UniProt database (https://www.uniprot.org, 8 February 2022), and the three dimensional structure of the protein was downloaded from the protein database (https://www.rcsb.org, 8 February 2022). The two dimensional structure of IQ was downloaded from the PubChem database (https://pubchem.ncbi.nlm.nih.gov, 5 February 2022). The target protein was pretreated by removing solvent molecules, inorganic sub-receptors, hydrogenated, charged, and active components as ligands, as well as the grid box coordinates and sizes based on the target protein. Finally, the file with the lowest free binding energy was selected and exported to “.pdbqt” format, then converted to “.pdb” format, and imported into PyMol software for visualization.

### 4.9. Statistical Analysis

All results were expressed as means ± SD of three independent experiments. Statistical analyses were performed using one-way ANOVA by GraphPad Prism 8 software. * *p* < 0.05 was considered as significant.

## 5. Conclusions

The results of this study confirmed the activity of IQ in antioxidative damage and the anti-apoptosis of MC3T3 and further verified the effectiveness of IQ in promoting the osteogenic development of zebrafish. Based on the level of protein expression, IQ exhibited a disturbance in the Keap1-Nrf2-ARE signal pathway and mitochondria-mediated Caspase3 protein pathway, and IQ may play a regulatory role by binding to Keap1 protein. In the follow-up research, more animal and mechanism studies should be conducted to provide a more theoretical basis for the development of IQ as a natural supplement to promote bone development and repair.

## Figures and Tables

**Figure 1 molecules-27-03459-f001:**
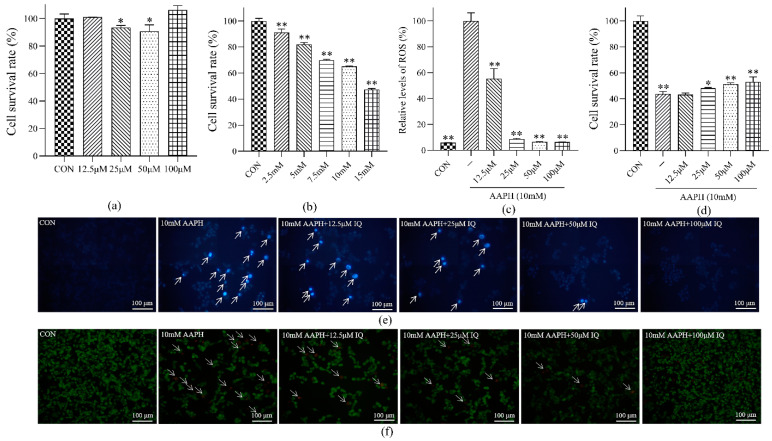
Effect of IQ on AAPH-induced viability and apoptosis of MC3T3 cells. (**a**) Cytotoxicity experiment of IQ on MC3T3 cells; (**b**) concentration screening experiment of AAPH on MC3T3 cells oxidative damage; (**c**) detection experiment of IQ on the relative level of ROS in MC3T3 cells induced by AAPH; (**d**) protective experiment of IQ on AAPH-induced MC3T3 cells oxidative damage and apoptosis; (**e**) staining of apoptotic bodies in the MC3T3 cell nucleus (Hoechst 33342); (**f**) staining experiment of MC3T3 during nuclear apoptosis (AO/EB). Experiments were performed in triplicate, and the data were expressed as the mean ± SD, * *p* < 0.05, ** *p* < 0.01.

**Figure 2 molecules-27-03459-f002:**
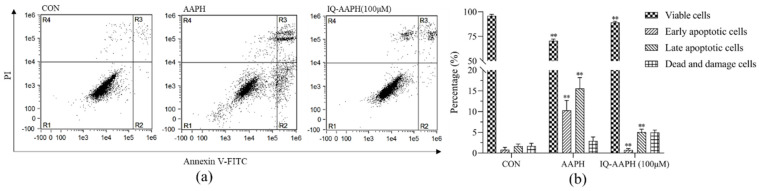
Effect of IQ on AAPH-induced apoptosis cycle of MC3T3 cells. (**a**) Apoptosis detected by flow cytometry; (**b**) statistical analysis of cell proportion in different stages of apoptosis. Experiments were performed in triplicate, and the data were expressed as the mean ± SD, ** *p* < 0.01.

**Figure 3 molecules-27-03459-f003:**
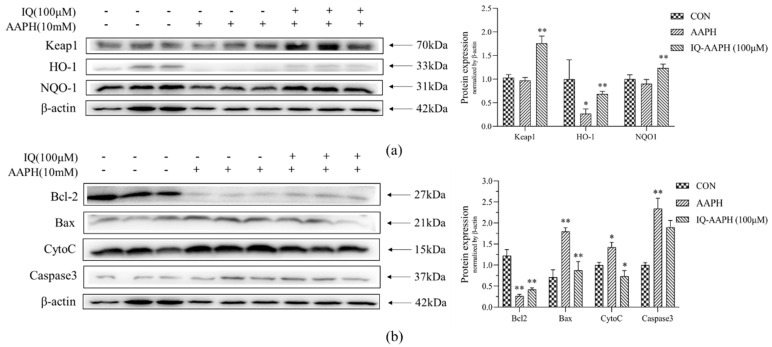
Effect of IQ on AAPH-induced protein expression of MC3T3 cells. (**a**) Protein expression levels of the CON group, AAPH group, and IQ-AAPH group on the Keap1-Nrf2-ARE signaling pathway; (**b**) the protein expression levels of three groups on the Caspase3 signaling pathway. Experiments were performed in triplicate, and the data are expressed as the mean ± SD, * *p* < 0.05, ** *p* < 0.01.

**Figure 4 molecules-27-03459-f004:**
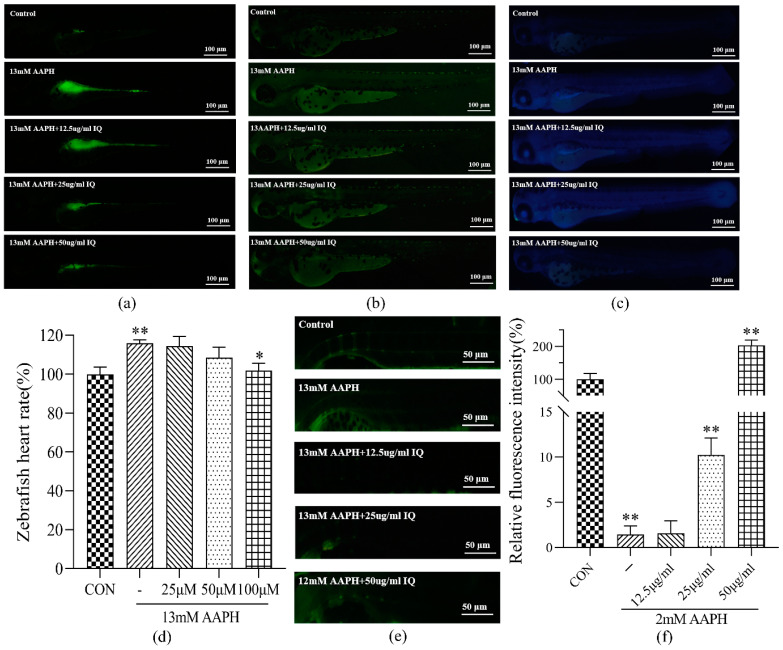
Effect of IQ on AAPH-induced antioxidant and injury repair of zebrafish. (**a**) ROS staining experiment of 72 hpf zebrafish; (**b**) apoptotic cell staining experiment of 72 hpf zebrafish; (**c**) lipid peroxidation staining experiment of 72 hpf zebrafish; (**d**) heart rate measurement of zebrafish; (**e**) staining experiment of 9 dpf zebrafish osteogenesis; (**f**) fluorescence intensity analysis of spine mineralization in 9 dpf zebrafish. Experiments were performed in triplicate, and the data were expressed as the mean ± SD, * *p* < 0.05, ** *p* < 0.01.

**Figure 5 molecules-27-03459-f005:**
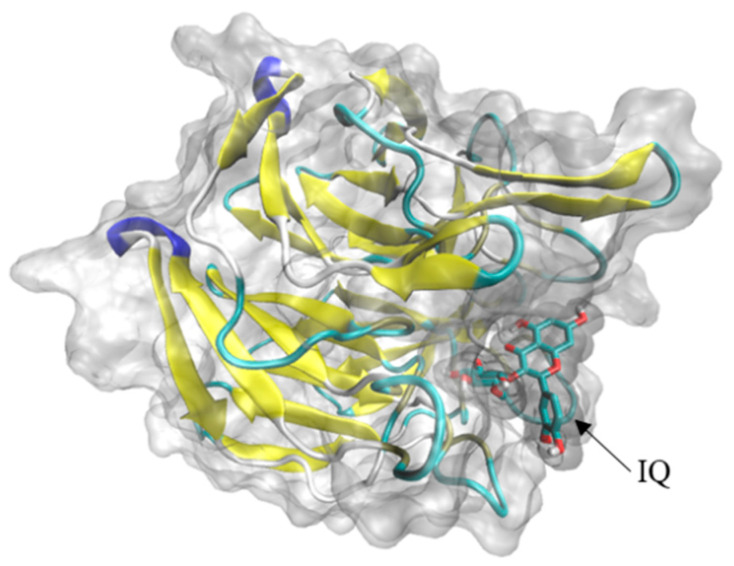
3D molecular docking diagram of IQ with Keap1. Affinity = −8.6 kcal/mol. Coordinate parameter: center x = 2.102, center y = 31.44, center z = 68.757, size x = 59.85, size y = 51.3, size z = 44.65; other parameters are the default Auto Dock Tools parameters.

**Table 1 molecules-27-03459-t001:** Binding energy of IQ and Keap1.

Mode	Affinity (kcal/mol)	Dist from rmsd l.b	Best Mode rmsd u.b
1	−8.600	0.000	0.000
2	−8.500	1.075	1.430
3	−8.400	1.689	2.728
4	−8.200	2.787	5.201
5	−8.200	2.257	3.592
6	−8.000	2.045	6.699
7	−7.800	1.963	5.287
8	−7.700	3.241	5.007
9	−7.500	21.886	25.630

## Data Availability

Not applicable.

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
