# Peer review of "Isoquercitrin Attenuates Osteogenic Injury in MC3T3 Osteoblastic Cells and the Zebrafish Model via the Keap1-Nrf2-ARE Pathway"

_molecules, 2022, doi:10.3390/molecules27113459_

Round 1
Reviewer 1 Report
This work observed the anti-oxidative and anti-apoptosis activities of IQ in MC3T3 cells and further verified the effectiveness of IQ in promoting the osteogenic development of zebrafish. Based on the level of protein expression, IQ had a disturbance in the Keap1-Nrf2-ARE signal pathway and mitochondria-mediated Caspase3 protein pathway, and IQ may play a regulatory role by binding to Keap1 protein. These results are convincing, clear and publishable. Best of luck, keep doing this type of novel research work. However, I request the following clarifications or modification.
- Solvents in IQ and AAPH solution used in cells and zebrafish model should be explained. The living environment of zebrafish embryos is E3 water. How is E3 water obtained? Please explain the configuration method.
- When screening AAPH concentration, the gradient settings for different concentrations were relatively centralized, which is not convincing. Please explain the reason.
- Line 361-362: “Separately, 200 μl of cell suspension in the CON group, AAPH group, and IQ-AAPH group were mixed with 1.5 ml ep tube” This statement may be unbefitting.
- Line 385: “The membrane was successfully transferred by Ponceau red staining.” This statement is hardly understood.
- The relationship between oxidative damage and bone damage is not explained in the text. Please try to explain clearly.
- When performing molecular docking, did the protein on both pathways undergo simulated docking? How to conclude that IQ plays a role by specifically binding to Keap1?
- What is the direct relationship between the antioxidant effect of IQ on 72hpf zebrafish and the bone injury repair effect of IQ on 9dpf zebrafish.
- In Discussion sections, I recommended to explain the connection between the two pathways: Keap1-Nrf2-ARE signaling pathway and Caspase3 signaling pathway.
Author Response
Thank you very much for assessing our manuscript. We appreciate your detailed review and comments. We highlighted the revised parts of the manuscript in blue colours.
Please see the attachment.

Reviewer 2 Report
Isoquercitrin (IQ) has well-established potent antioxidant properties. Its beneficial effects in bone metabolic bone diseases has also been widely reported, along with a variety of other conditions where oxidative stress plays a role. Additionally, the involvement of the Keap1-Nrf2-ARE pathway in this protective mechanism has also been reported. In this way, I think this article might be improved in several aspects, for a better understanding and to catch the interest of the less informed readers in this issue.
Introduction:
- Role of oxidative stress in bone metabolic diseases, particularly osteoporosis.
- Provide a brief general view of the role of the Keap1-Nrf2-ARE pathway in protecting against oxidative stress injury under physiological and pathological conditions, particularly osteoporosis (as it was the target aim of this study).
- Provide a more detailed explanation of the anti-oxidant mechanism involved in the protective effect of Isoquercitrin, via the Keap1-Nrf2-ARE pathway.
- Provide a brief information and relevance on the molecule 2,2-azobis(2-amidinopropane) dihydrochloride (AAPH) as a model to address oxidative stress-related studies.
- Further and clearly evidence the added value of this study, compared to the state of the art.
Results:
- Provide better resolution fluorescence images (Figs. 1 and 4).
Discussion:
- Provide a brief comparison of the present results concerning bone cells/bone tissue to other studies also showing the beneficial effect of Isoquercitrin in other oxidative stress-induced conditions and the involvement of Keap1-Nrf2-ARE pathway.
Author Response
Thank you very much for assessing our manuscript. We appreciate your detailed review and comments. We highlighted the revised parts of the manuscript in green colours.
Please see the attachment.

Reviewer 3 Report
See attachment.

Author Response
Thank you very much for assessing our manuscript. We appreciate your detailed review and comments. We highlighted the revised parts of the manuscript in red colours.
Please see the attachment.

Round 2
Reviewer 2 Report
The authors addressed my sugestions, improving the quality of the manuscript.
Reviewer 3 Report
I have not received a satisfactory response from the authors. I believe that research should include whether previous studies are correct. I do not think the authors' protocol satisfies the objective.